# A target tracking method based on adaptive occlusion judgment and model updating strategy



Zhiming Cai[1,2,*], Zhuangzhuang Wang[1,*], Jianchao Huang[1], Shujing Chen[1] and Huabin He[1]

[1] School of Electronic, Electrical Engineering and Physics, Fujian University of Technology, Fuzhou, Fujian, China
[2] National Demonstration Center for Experimental Electronic Information and Electrical Technology Education, Fujian University of Technology, Fuzhou, Fujian, China
* These authors contributed equally to this work.

## ABSTRACT

Target tracking is an important research in the field of computer vision. Despite the rapid development of technology, difficulties still remain in balancing the overall performance for target occlusion, motion blur, *etc.* To address the above issue, we propose an improved kernel correlation filter tracking algorithm with adaptive occlusion judgement and model updating strategy (called Aojmus) to achieve robust target tracking. Firstly, the algorithm fuses color-naming (CN) and histogram of gradients (HOG) features as a feature extraction scheme and introduces a scale filter to estimate the target scale, which reduces tracking error caused by the variations of target features and scales. Secondly, the Aojmus introduces four evaluation indicators and a double thresholding mechanism to determine whether the target is occluded and the degree of occlusion respectively. The four evaluation results are weighted and fused to a final value. Finally, the updating strategy of the model is adaptively adjusted based on the weighted fusion value and the result of the scale estimation. Experimental evaluations on the OTB-2015 dataset are conducted to compare the performance of the Aojmus algorithm with four other comparable algorithms in terms of tracking precision, success rate, and speed. The experimental results show that the proposed Aojmus algorithm outperforms all the algorithms compared in terms of tracking precision. The Aojmus also exhibits excellent performance on attributes such as target occlusion and motion blur in terms of success rate. In addition, the processing speed reaches 74.85 fps, which also demonstrates good real-time performance.

## INTRODUCTION

Motion target tracking (*Lu & Xu, 2019*; *Wang et al., 2021*) is one of the most active research areas in computer vision. With the continuous improvement of hardware facilities and the rapid development of artificial intelligence technology, motion target tracking technology is widely used in intelligent video surveillance (*Zeng et al., 2020*),

Corresponding author
Zhiming Cai, caizm@fjut.edu.cn

human-computer interaction (*Zhou & Liu, 2021*), medical diagnosis (*Al-Battal et al., 2021*) and other fields. In the field of intelligent video surveillance, target tracking technology is commonly used in monitoring of vehicle violations and has been proved to be effective. In the field of medical diagnosis, tracking technology is frequently used in tracking microscopic items like cells. In terms of human-computer interaction, tracking technology is mostly utilized in robot vision and virtual environments, which primarily use visual technology to provide the tracking effect similar to human eyes. In the past two decades, target tracking technology has made tremendous developments. However, tracking targets are often limited by complex application environments, such as different illumination changes, interference from complex backgrounds, changes in their own scales, and occlusion by other objects. Therefore, improving the precision and robustness of tracking algorithms in complex environments and satisfying real-time applications become important research topics in visual target tracking.

Nowadays, the mainstream algorithms of target tracking can be classified into two categories. One is based on correlation filtering (*Wei & Kang, 2017*; *Meng & Li, 2019*), which determines the correlation region by establishing a correlation filter to find the maximum response value in the two adjacent frames, and then lock the target. Compared with the earlier tracking algorithms based on optical flow method (*Xiao et al., 2016*) and feature matching (*Uzkent et al., 2015*), the advantages for this category are fast speed and good robustness in the case of target occlusion, illumination change and motion blur (*Liu et al., 2017*). Another is based on deep learning (*Li, Li & Porikli, 2016*), which uses convolutional neural networks training to extract object features in the last frame and matches the object in the next frame. That is, the object is continually tracked during training. For one thing, the former is inferior to the latter when dealing with complex scenarios, such as target occlusion, out-of-view, scale variation *etc*. For another, the latter tracks object more slowly. Therefore, finding a solution not only meets the demands of accurate tracking in a variety of complex scenarios, but also achieves fast running are still an active research area. In this work, a target tracking algorithm based on adaptive occlusion judgment and model updating strategy, called Aojmus, is proposed to address the poor tracking performance in complex scenarios mentioned above. The Aojmus is designed on the basis of KCF algorithm and integrated with correlation filtering method which has the advantage in processing speed.

The contributions of this work can be summarized in three folds:

1. We propose to fuse CN and HOG features as the feature representation of the tracked target, which improves the discrimination and re-detection ability. Meanwhile, the scale filter is introduced to solve the defect of poor tracking precision of the target due to scale change.

2. We design four kinds of occlusion judgment indicators to solve tracking failure which caused by occlusion. These four indicators can adaptively judge the occlusion of the target during tracking. A double threshold mechanism is introduced to judge the degree of occlusion, which determines the update strategy of the tracker.

3. We use a weighted fusion strategy to fuse the results of occlusion judgments to ensure that the model update rate of the tracker changes dynamically with the judgment results of each frame, avoiding the tracking drift problem caused by fixed model update rate of most trackers.

The rest of this article is organized as follows. In "Literature Review", related works about target tracking are surveyed. In "Preliminaries", some prerequisites of the methodology are introduced. In "Methodology", we describe the architecture of the proposed algorithm, including statistical analysis, algorithm design and related parameter setting. In "Experiments and analysis", we compare and analyze the performance of the algorithm in quantitative and qualitative aspects. In "Conclusion", we summarize this study and discuss possible future work.

## LITERATURE REVIEW

*Bolme et al. (2010)* were the first to apply the correlation filtering method to target tracking and proposed the MOSSE algorithm, which achieves tracking speed of 669 fps, but with a slightly poor precision of 43.1%. To solve the problem of insufficient samples of MOSSE algorithm, *Henriques et al. (2012)* proposed CSK algorithm, which acquired a large number of samples through the method of cyclic shift. Moreover, the computational complexity is reduced by frequency domain processing, and thereby a robust and accurate filter is obtained. Subsequently, *Henriques et al. (2015)* proposed the kernelized correlation filter (KCF) tracking algorithm on the basis of CSK, which utilized the histogram of oriented gradients (HOG) feature instead of grayscale feature and introduced a circular matrix to reduce the computational effort. The algorithm also incorporates multi-channel data to improve the operation speed and meet the requirement of real-time in the process of tracking. Inspired by the scale pooling technique, *Yang & Zhu (2014)* and *Danelljan et al. (2014a)* proposed SAMF and DSST algorithms respectively, which solved the problem of scale adaptation of KCF algorithm. The SAMF algorithm fused HOG feature and color-naming (CN) (*Danelljan et al., 2014b*) feature for the first time on the basis of KCF, which improved the tracking precision, but the speed is significantly reduced. Similarly, the DSST algorithm also achieved scale adaption, but the overall performance is inferior. *Danelljan et al. (2017a)* used convolutional neural network (CNN) to extract depth features on the feature model of the target while keeping the motion model (cyclic matrix) and observation model (correlation filter) unchanged, achieving a significant increase in precision and success rate. The research interest in correlation filtering-based target tracking has declined because the precision is difficult to improve further when dealing with target occlusion, disappearance, or non-rigid object tracking. In recent years, some representative studies have still emerged, such as ASRCF (*Dai et al., 2019*), ARCF (*Huang et al., 2019*), and PRCF (*Sun et al., 2019*).

Another class of target tracking algorithms is based on deep learning (*Li, Li & Porikli, 2016*), which uses convolutional neural networks for feature extraction and classification of targets to achieve target tracking. Some of them incorporate correlation filtering and deep

learning, such as the HCF (*Ma et al., 2015*). The MDNet algorithm proposed by *Nam & Han (2016)* was one of the early algorithms that used deep learning alone to implement target tracking. The algorithm trains each domain separately while updating the parameters of the shared layer during training so that these parameters can be adapted to all datasets. When tracking, MDNet uses a pre-trained CNN network to track the target and thereby locate the target. Here are some similar algorithms, such as SiamDW (*Zhang & Peng, 2019*), SiamCAR (*Guo et al., 2020*) and HiFT (*Cao et al., 2021*), *etc.* Despite the superior performance of deep learning-based tracking algorithms in achieving tracking precision, they still face the disadvantages of insufficient initial training samples and slow tracking speed.

## PRELIMINARIES

In this section, to help establish an understanding of the essential elements involved in the proposed methodology, the kernelized correlation filter (KCF), classifier training, fast detection and model updating are illustrated in advance.

### Kernelized correlation filter

The core idea of the kernel correlation filtering (KCF) algorithm is to calculate the matching degree between the predicted region and the target by establishing a kernel function based on the ridge regression. By moving the complex calculation to the frequency domain with fast Fourier transform, the fast tracking for target is achieved. Similar to most discriminative tracking algorithms, KCF algorithm also performs target detection before filter model training. It firstly trains a model of the initial position of the target, then detects whether the target exists in the prediction region of the next frame, and finally uses Gaussian kernel to calculate the correlation between two adjacent frames and determines the position of the target according to its maximum response value in the target region. The basic principles of the kernel correlation filtering algorithm, including classifier training, fast detection and model update, are described below.

### Classifier training

The classifier $f(x) = \langle w, \varphi(x) \rangle$ is obtained by training ridge regression. Let $(x_i, y_i)$ be the training sample, where $y_i$ is the regression expectation corresponding to sample $x_i$. The ridge regression on the training sample yields the linear regression function $f(x) = w^T x_i$. To prevent the overfitting phenomenon, the classifier needs to be regularized as follows:

$$\min_{w} \sum_{i=1}^{N} (f(x_i) - y_i)^2 + \lambda \| w \|^2 \tag{1}$$

where $w$ is the classifier parameter and $\lambda$ is the regularization parameter. The closed-form solution of the above equation is:

$$w = (X^T X + \lambda I)^{-1} X^T y \tag{2}$$

In the process of generating a large amount of information of target and background using the circular matrix, the feature space formed by the sample set appears nonlinear.

Therefore, the Gaussian kernel function $\varphi(x_i)$ is introduced for linear transformation, and the result is $f(x) = w^T x_i = w^T \varphi(x_i)$, where $w = \sum_{i=1}^{N} \alpha_i x_i$. So far, the solution of $w$ is transformed to the solution of coefficient $\alpha$, which eventually yields:

$$\alpha = (K + \lambda I)^{-1} y \tag{3}$$

where $K$ is the kernel correlation matrix. To reduce the complexity of the calculation, Eq. (3) is transformed into the frequency domain with the discrete Fourier transform (DFT). Then the solution becomes:

$$\hat{\alpha} = \frac{\hat{y}}{k^{xx} + \lambda} \tag{4}$$

The purpose of classifier training is to solve the weight coefficient $\alpha$, where $k^{xx}$ is the first row element of the kernel cycle matrix $K$, and $\hat{}$ denotes the DFT of vector.

## Fast detection and model updating

After classifier training, in order to locate the target position of the current frame, the KCF algorithm uses the target position of the previous frame as a template, then detects it in the candidate region $z$ of the current frame and determines the target position by finding the maximum value of $f(z) = \alpha^T \varphi(X) \varphi(z)$. To increase the calculation speed, the KCF algorithm transfers the solution from the time domain to the frequency domain as follows:

$$\hat{f}(z) = \hat{k}^{xz} \odot \hat{\alpha} \tag{5}$$

where $k^{xz}$ denotes the kernel correlation between the target sample $x$ and the candidate detection region $z$, $\hat{f}(z)$ represents the response distribution in the candidate region and the position where its maximum value is located indicates the actual position of the target in the current frame.

To ensure each frame in the video sequence can be processed, the KCF algorithm uses linear interpolation to update the filter template $\hat{\alpha}_t$ and the target feature template $\hat{x}_t$ as follows:

$$\begin{cases} \hat{\alpha}_t = (1 - \eta) \hat{\alpha}_{t-1} + \eta \hat{\alpha} \\ \hat{x}_t = (1 - \eta) \hat{x}_{t-1} + \eta \hat{x} \end{cases} \tag{6}$$

where $\eta$ denotes the model update rate and $t$ is time stamp.

## METHODOLOGY

As target occlusion, scale variation, illumination variation affect the performance of tracking, it is of great significance to conquer such problems. The KCF algorithm increases the training samples through the circular matrix, which in turn improves the tracking accuracy. Meanwhile, by transferring to the frequency domain to avoid matrix inversion operations, the computation is greatly reduced.

However, the KCF algorithm often fails to track in the case of target occlusion or target loss because the update model learns the features of the occluded object and causes the model to get the wrong target features in the accumulation of subsequent frames, which in

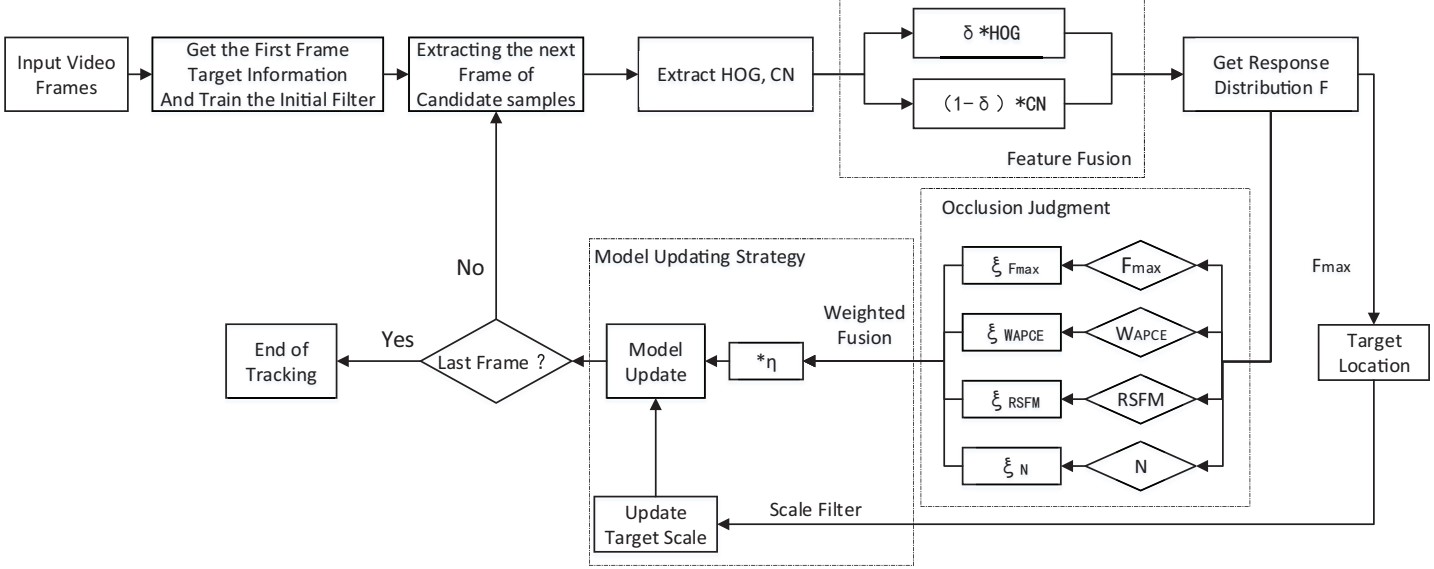

**Figure 1 Flow chart of Aojmus algorithm.**

turn leads to tracking failure. In addition, the tracking box of KCF algorithm cannot meet the scale variation of the target, which can also greatly reduce the precision of tracking. To address these problems, this article proposes a target tracking algorithm, call Aojmus, based on an adaptive occlusion judgment and model update strategy. The flow chart of the algorithm is shown in Fig. 1.

In this section, we describe the specific implementation of Aojmus, including feature fusion, scale estimation, occlusion judgment and model updating.

## Feature fusion design

The HOG features can effectively depict the local contour and shape information of the target and are very robust to illumination changes, but are poorly adapted to target deformation and fast motion. The CN features can well represent the global color information of the target and have excellent stability to target deformation and fast motion, but are sensitive to illumination and color changes. Therefore, we employ linear fusion of HOG feature and CN feature (*Xie & Zhao, 2021*) to achieve feature complementarity and improve tracking precision.

The process of linear weighted fusion of these two feature vectors is as follows:

$$v_{hc} = \delta v_{\text{hog}} + (1 - \delta)v_{cn} \tag{7}$$

where $v_{hog}$, $v_{cn}$, $v_{hc}$ represent HOG feature, CN feature and fused feature respectively, and $\delta$ is the weighted coefficient of feature fusion. In this article, set $\delta = 0.5$ to ensure that the advantages of HOG and CN feature can be fully utilized.

## Multi-scale estimation

The scale variation of target is one of the important factors affecting the tracking results. As the position change of two consecutive frames is often larger than the scale change, like DSST (*Danelljan et al., 2014a*), this article first uses a two-dimensional position filter to determine the position information and then implements scale evaluation by training a one-dimensional scale filter.

Let $f$ be the training sample and $h$ be the optimal correlation filter. The minimum cost function is solved with ridge regression as follows:

$$\varepsilon = \left\| \sum_{l=1}^{d} h^l \odot f^l - g \right\|^2 + \lambda \sum_{l=1}^{d} \| h^l \|^2 \tag{8}$$

where $l \in \{1, \ldots, d\}$ is the feature dimension, $g$ represents the regression expectation corresponding to the training sample $f$, and $\lambda$ is the regularization factor. The scaling filter can be obtained by solving the above equation in Fourier domain:

$$H^l = \frac{\bar{G}F^l}{\sum_{k=1}^{d} \overline{F^k}F^k + \lambda} = \frac{A_t^l}{B_t} \tag{9}$$

where $\bar{G}$ represents complex conjugate of the DFTs of correlation outputs and $\lambda$ is introduced to avoid zero denominator in case of the zero frequency component in $f$. By detecting the image block $z$ in the new frame, we can obtain the response of scale filter as:

$$y = F^{-1} \left\{ \frac{\sum_{l=1}^{d} \overline{A^l} Z^l}{B + \lambda} \right\} \tag{10}$$

Up to this point, the response value of the scale filter can be calculated from Eq. (10), and a new scale estimate can be determined based on the result of the maximum value. The selection principle of target sample size for scale evaluation is as follows:

$$a^n P \times a^n R, n \in \left\{ \left\lfloor \frac{-(s-1)}{2} \right\rfloor, \ldots, \left\lfloor \frac{(s-1)}{2} \right\rfloor \right\} \tag{11}$$

where $P$ and $R$ are respectively the width and height of the target in the previous frame, $a = 1.02$ is the scale factor, $s = 33$ is the length of the scale filter.

## Adaptive occlusion judgment and model updating strategy

Target occlusion often occurs during tracking. In this section, we make a detailed analysis and propose an adaptive judgment method. Using the original model update rate in the KCF algorithm, the response of occlusion is analyzed with the FaceOcc1 image sequence in the OTB2015 (*Wu, Lim & Yang, 2015*) dataset as an example.

From the response distribution in Fig. 2, it can be seen that the main peak of the response in target box during tracking without occlusion is dominated and there is no other obvious peaks. The maximum peak response is close to 1. When the occlusion exists, as shown in Fig. 3, the maximum peak response decreases significantly, and the rest of the

Frame 17

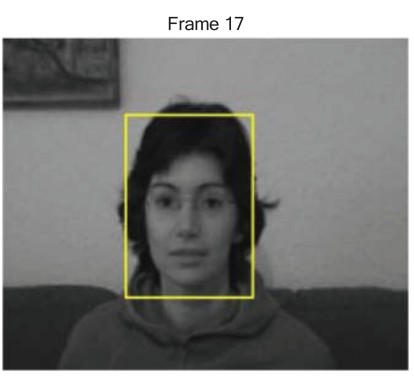

Response of frame 17

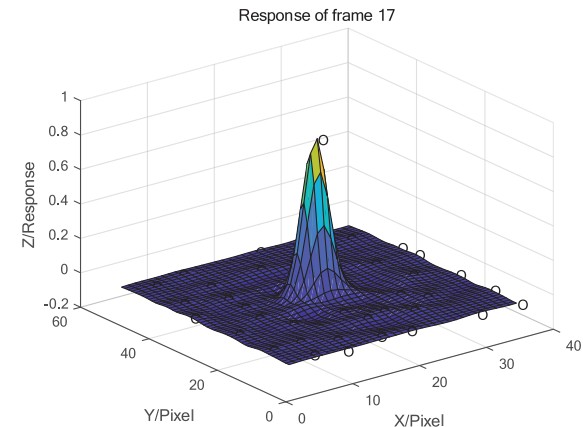

**Figure 2 Tracking results and response distribution at frame 17 of the FaceOcc1 sequence without occlusion.** Photo credit: Visual Tracker Benchmark.

Frame 91

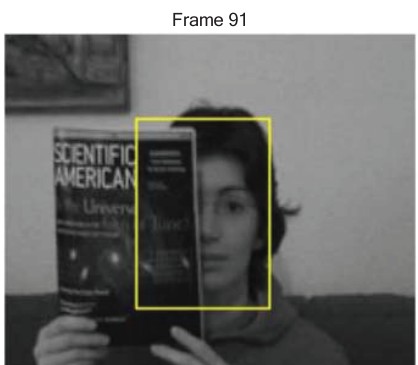

Response of frame 91

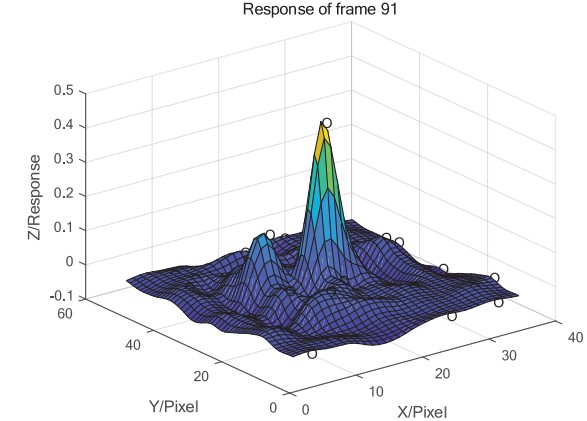

**Figure 3 Tracking results and response distribution when occlusion appears at frame 91 of FaceOcc1 sequence.** Photo credit: Visual Tracker Benchmark.

peak responses increase and become more prominent. It can be concluded that when the occlusion occurs, the fixed model update strategy learns the features of the occlusion and applies this feature to the search of the next frame which leading to the appearance of other peaks besides the main peak. Hence, the presence of occlusion can be determined based on the response distribution of the target.

Let $F_{max}$ be the maximum peak response in each frame. The number of peak response points that exceed the maximum peak response by a certain proportion, denoted as $N$, can be expressed as:

$$N = \sum \left( \left( F'_{\max} > \zeta F_{\max} \right) \in S \right) \qquad (12)$$

where $F'_{max}$ denotes the peak response other than the maximum peak, $S$ represents the target area, and $\zeta$ is a proportionality coefficient which is set to 0.1 in this article.

Besides the two judgment indicators of $F_{max}$ and $N$, two more judgment indicators, average Peak-to-Correlation energy (APCE) (*Wang, Liu & Huang, 2017*) and ratio between the second and first major mode (RSFM) (*Lukevic et al., 2017*) are introduced in this article to ensure the robustness of the occlusion judgment.

The APCE can well reflect the variation of response and changes significantly when the target is obscured. It can be expressed as follows:

$$W_{APCE} = \frac{|F_{\max} - F_{\min}|^2}{mean\left(\sum_{w,h}\left(F_{w,h} - F_{\min}\right)^2\right)} \qquad (13)$$

where $F_{min}$ denotes the minimum value of the response, and $F_{w,h}$ denotes the response value of pixel in $w$-th row and $h$-th column. When occlusion appears, the value of $W_{APCE}$ will decrease significantly.

The RSFM reflects the prominence of the main peak in the response map and is defined as follows:

$$RSFM = 1 - \min\left(\frac{F_{\text{second}}}{F_{\max}}, \frac{1}{2}\right) \qquad (14)$$

where $F_{second}$ represents the response value of the second peak. The larger the value of RSFM is, the more prominent the main peak will be and the higher reliability the tracking will have, and *vice versa*.

In this article, we use the four evaluation indicators mentioned above, $F_{max}$, $N$, $W_{APCE}$ and RSFM, to determine whether the target is occluded or not. To verify the four indicators, we select the first 200 frames of the video sequence in FaceOcc1 for test. The relevant results of these four indicators for each frame are calculated and their cumulative averages are shown in Fig. 4.

As shown in Fig. 4, the evaluation indicators fluctuate significantly between frame 85 to 100. The $F_{max}$, $W_{APCE}$, RSFM are relatively low whereas $N$ is relatively high, which indicates the existence of severe occlusion. By comparing the moments when occlusion appears in the original video sequence, it can be found that the above four indicators satisfy well for the judgment of occlusion as they are complementary to some extent.

In order to accurately determine whether occlusion exists and the degree of occlusion, this article implements adaptive evaluation by setting dynamic double thresholds. The thresholds are selected based on the average value of each evaluation indicator for the previous $t - 1$ frames, namely:

$$\theta(R)_n = \frac{\kappa_n}{t-1}\sum_{i=2}^{t}R^i, R \in \left(F_{\max}, W_{APCE}, RSFM, N\right), n = 1, 2 \qquad (15)$$

where, $\kappa$ denotes the weighted coefficient, and $\theta(R)$ denotes the threshold of the corresponding evaluation indicator. For every indicator, there are two thresholds. The

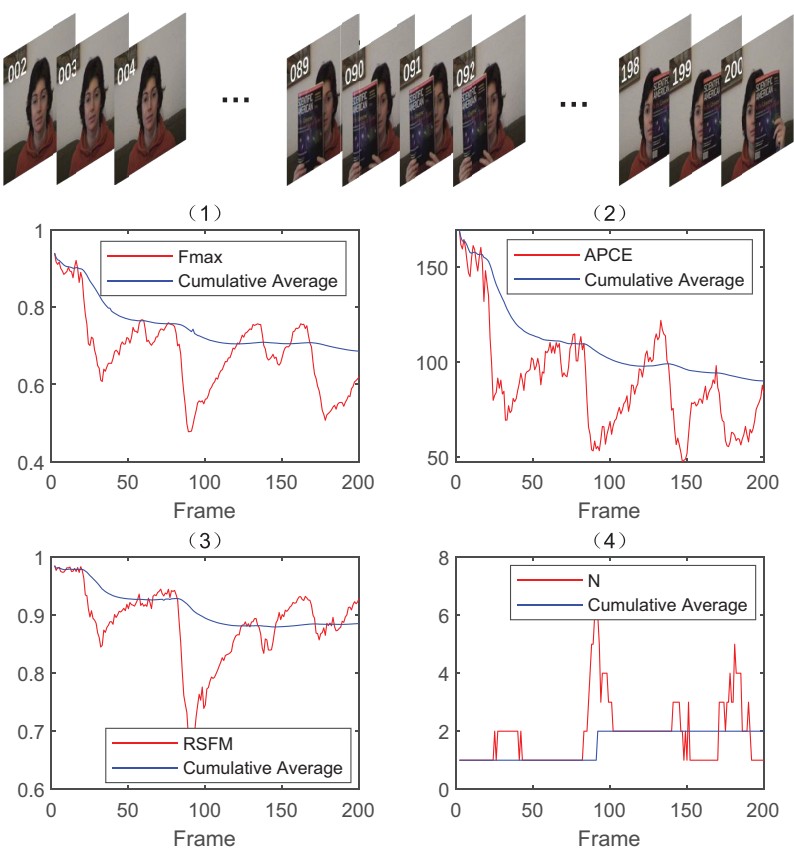

**Figure 4 Line graphs of $F_{max}$, $W_{APCE}$, $RSFM$ and $N$ for the front 200 frames of FaceOcc1 as an example.** Photo credit: Visual Tracker Benchmark.

**Table 1 Weighted coefficients of double thresholds for each evaluation indicator.**

|  | $\kappa_1$ | $\kappa_2$ |
|---|---|---|
| $\theta(F_{max})_n$ | 1 | 0.85 |
| $\theta(W_{APCE})_n$ | 1 | 0.7 |
| $\theta(RSFM)_n$ | 1 | — |
| $\theta(N)_n$ | 1 | — |

weighted coefficient $\kappa$ of each evaluation indicator is obtained by analyzing the cumulative average curve in Fig. 4, as shown in Table 1.

As can be seen from Eq. (14), the value of $F_{second}$ rises sharply when there is severe occlusion in current frame, resulting the output of $RSFM$ to be 0.5. When it occurs, the lower limit of $\theta(RSFM)_2$ in Table 1 is set to 0.5 particularly. As $N$ is usually not sensitive to occlusion, it is set to a single threshold for simplicity.

When occlusion appears, the fixed model update rate $\eta$ will cause the tracker to learn the information of the occlusion object and lead to tracking drift. In this article, the Aojmus algorithm adaptively generates a suitable model update rate according to the degree of

**Table 2 Output values of the four judgment indicators $\mu_n$.**

|  | $\mu_1$ | $\mu_2$ | $\mu_3$ |
|---|---|---|---|
| $\xi_{F_{max}}$ | 1 | 1.2 | 1.5 |
| $\xi_{W_{APCE}}$ | 1 | 0.8 | 0.5 |
| $\xi_{RSFM}$ | 1 | 1.2 | 1.5 |
| $\xi_N$ | 1.2 | — | 1 |

---

**Algorithm 1 Aojmus**

**Input:**

    Current frame $I_t$; The video for tracking $S_{videos}$;

    Target position $p_{t-1}$ and scale $s_{t-1}$ of previous frame;

    The target feature template, $\hat{x}_{t-1}$ and the filter template, $\hat{\alpha}_{t-1}$;

    The scale model $A_{t-1}^{scale}$, $B_{t-1}^{scale}$.

**Output:**

    Target position $p_t$ and scale $s_t$ of current frame;

    The updated target feature template, $\hat{x}_t$ and filter template, $\hat{\alpha}_t$;

    The updated scale model $A_t^{scale}$, $B_t^{scale}$.

1:  **for** each $I_t \in S_{videos}$ **do**

2:     Sample the new patch $z^t$ from $I_t$ at $p_{t-1}$;

3:     Extract a scale sample $z_{scale}$ from $I_t$ at $p_t$ and $s_{t-1}$;

4:     Extract the HOG and CN features and fused with Eq. (7);

5:     Calculate the response $\hat{f}(z^t)$ with Eq. (5), and get $F_{max}$;

6:     Calculate $N$, $W_{APCE}$ and $RSFM$ with Eqs. (12)–(14), and adaptively judge whether there is occlusion and the scope;

7:     Get the $\xi_R$ and $\xi$ by Eqs. (16) and (17);

8:     Compute the scale correlations $y_{scale}$ using $z_{scale}$, $A_{t-1}^{scale}$ and $B_{t-1}^{scale}$ in Eq. (10);

9:     Set $s_t$ to the maximum of $y_{scale}$;

10:    Use Eq. (18) to update $\hat{x}_t$ and $\hat{\alpha}_t$ with $\hat{x}_{t-1}$ and $\hat{\alpha}_{t-1}$ adaptively;

11:    Use Eq. (9) to update $A_t^{scale}$ and $B_t^{scale}$ with $A_{t-1}^{scale}$ and $B_{t-1}^{scale}$.

12:    Return $p_t$ and the updated $\hat{x}_t$, $\hat{\alpha}_t$, $A_t^{scale}$, $B_t^{scale}$.

13:  **end for**

---

occlusion in current frame. When occlusion occurs, the model update rate is appropriately increased to ensure that the target information of the part not occluded in the tracking frame is fully learned by the model so as to enhance the model's recognition capability, which can be used for accurate localization in next frame. The update strategy is defined as:

$$\xi_R = \begin{cases} \mu_1 & R \geq \theta(R)_1 \\ \mu_2 & \theta(R)_1 > R > \theta(R)_2 \\ \mu_3 & else \end{cases} \qquad (16)$$

where, $R \in (F_{max}, W_{APCE}, RSFM, N)$, $\mu_n (n = 1, 2, 3)$ represent the output values of occlusion judgment. They are concluded from experiments, as shown in Table 2.

There are four judgment results in the improved algorithm. In order to guarantee that $\xi_R$ can accurately reflect the degree of occlusion, this article uses a weighted fusion of the four output $\xi_R$ to obtain the final model update rate as follows:

$$\xi = \frac{\xi_{F_{max}}^2 + \xi_{W_{APCE}}^2 + \xi_{RSFM}^2 + \xi_N^2}{sum(\xi_R)} \tag{17}$$

Finally, we substitute the final model update rate into Eq. (6) to obtain:

$$\begin{cases} \hat{\alpha}_t = (1 - \xi * \eta)\, \hat{\alpha}_{t-1} + \xi * \eta \hat{\alpha} \\ \hat{x}_t = (1 - \xi * \eta)\, \hat{x}_{t-1} + \xi * \eta \hat{x} \end{cases} \tag{18}$$

The proposed algorithm, Aojmus, is presented in Algorithm 1.

## EXPERIMENTS AND ANALYSIS

In this section the proposed Aojmus algorithm is compared with the other four relatively excellent tracking algorithms. We use three metrics to evaluate the performance of the algorithm, and select the representative video sequences to compare and analyze the tracking effect.

### Experimental environment and parameters

The platform for the experiments in this article is Matlab 2018a, and the hardware environment is a computer with Intel(R) Xeon(R) CPU E5-2620 v3 @ 2.40 GHz and 16 GB RAM. The parameters of the algorithm are set as follows: the regularization parameter $\lambda = 10^{-4}$ and the initial model update rate $\eta = 0.02$.

In this article, 66 video sequences provided by OTB-2015 are used for experimental verification. The dataset contains 11 attributes of common scenarios in target tracking, such as occlusion (OCC), deformation (DEF), illumination variation (IV), motion blur (MB), out-of-plane rotation (OPR), fast motion (FM), out-of-view (OV), in-plane rotation (IPR), low resolution (LR), scale variation (SV), and background clutter (BC). The performance evaluations are carried out with quantitative and qualitative analysis.

### Experimental comparison and analysis

#### Quantitative analysis

In order to evaluate the performance of the algorithm in this article, MSCF (*Zheng et al., 2021*), Staple (*Bertinetto et al., 2016*), fDSST (*Danelljan et al., 2017b*) and KCF algorithms with high performance were selected for comparison. Three statistical criteria of precision (*Pr*) (*Wu, Lim & Yang, 2013*), success rate (*Sr*) (*Wu, Lim & Yang, 2013*) and tracking speed (*Ts*) were used for evaluation respectively. The *Pr* refers to the error of center position, namely the Euclidean distance in pixel unit, $D_t$, between the center of tracking box for each

frame and the actual center in the benchmark. The final result is expressed with the average of errors.

$$Pr = \frac{D_t}{n} \tag{19}$$

The smaller the value of $Pr$ is, the closer the tracked target center to the actual location is and the better the algorithm performs in terms of pricision.

Let $O_t$ be the overlap between the tracking box in the current frame, $B_t$, and the actual box in benchmark, $B_{bt}$. It can be expressed as:

$$O_t = \frac{area(B_t \cap B_{bt})}{area(B_t \cup B_{bt})} \tag{20}$$

The success rate, $Sr$, is expressed as the average of $O_t$ whose value is greater than the given threshold in the whole video sequence.

$$Sr = \frac{1}{n}\sum_{t=1}^{n} O_t \tag{21}$$

The value of $Sr$ reflect the number of frames whose tracking box is closer to the real rectangle box. Obviously, the greater the $Sr$ is, the better the performance of the algorithm will be.

The tracking speed, $Ts$, refers to the number of video frames processed by the algorithm in each second, also known as frame rate with the unit of FPS (frames per second). It is defined as follows:

$$Ts = \frac{F_{total}}{t_{total}} \tag{22}$$

where, $F_{total}$ indicates the total number of frames of the video sequence, and $t_{total}$ is the time taken by the algorithm to run the whole video sequence.

To evaluate the performance of the proposed algorithm, Aojmus, we make a comparison with other four algorithms, MSCF, Staple, fDSST and KCF on OTB-2015 as shown in Figs. 5 and 6 and Tables 3–5. The evaluation method used is a one pass evaluation (OPE), which means that after initializing the target, the whole video sequence is run at once. The location error threshold for precision plots and the overlap threshold for success rate plots are set as 20 pixels and 0.5, respectively.

Figure 5 illustrates the precision and success rate of these five algorithms for running the whole sequences at once on the OTB-2015 dataset, from which the precision and success rate of Aojmus can be obtained are 0.909 and 0.749, respectively. Compared with others, the Aojmus performs the best in terms of precision and has improved 0.5% than the second ranked MSCF algorithm. Though the success rate is not outstanding, the Aojmus still shows high performance for OCC, MB, OV, and FM as shown in Fig. 6.

Tables 3 and 4 respectively show the precision and success rate of the five algorithms under 11 attributes. As it can be seen that the Aojmus algorithm performs best on 10 of

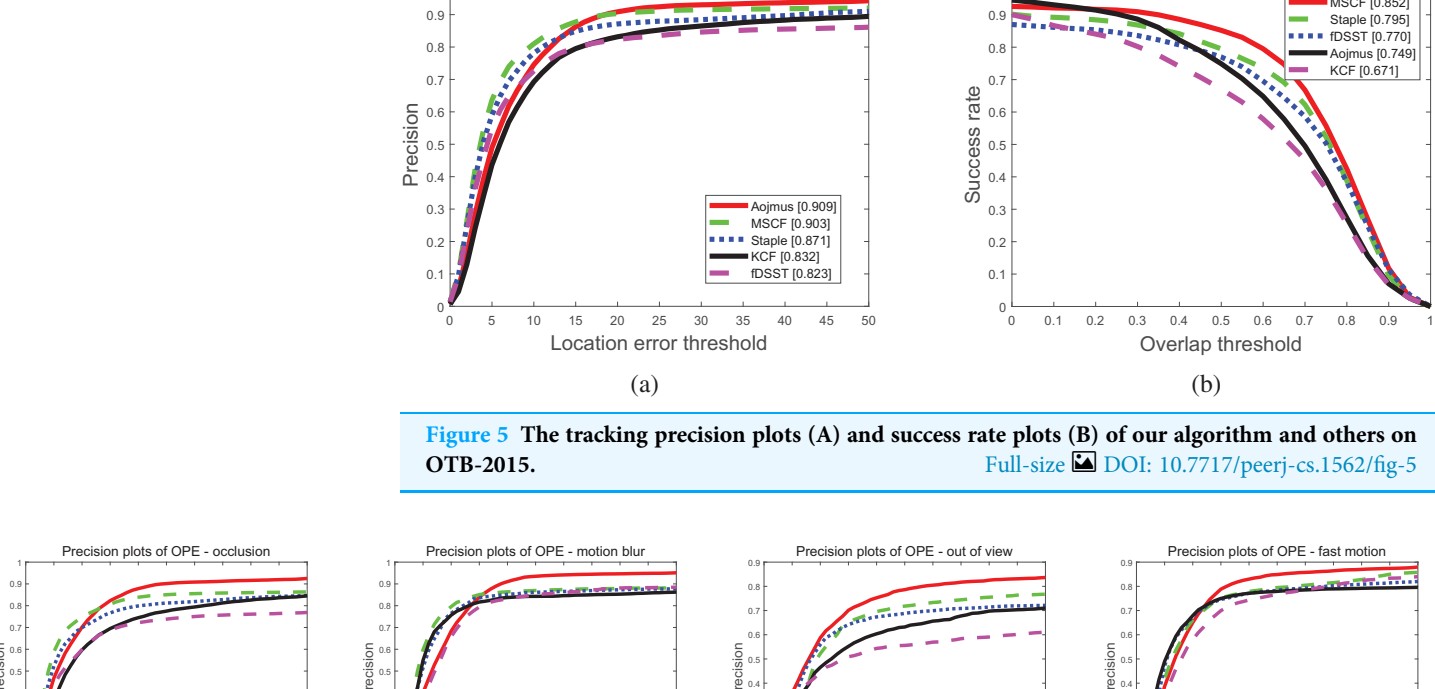

**Figure 5** The tracking precision plots (A) and success rate plots (B) of our algorithm and others on OTB-2015.

**Figure 6** Precision plots (above) and success rate plots (below) under OCC (A), MB (B), OV (C), FM (D).

these attributes. In terms of success rate, the Aojmus appears more robust in dealing with fast motion, motion blur and out of view problems. Despite the disadvantages in other aspects, the Aojmus is not much inferior to other excellent algorithms. For the tracking speed, though the Aojmus is inferior to fDSST and KCF as shown in Table 5, it outperforms others in other aspects as shown in Figs. 5 and 6. From the above comparisons, the Aojmus exhibits good overall performance.

**Table 3 Comparison of precision on different attributes.**

| Algorithms | OCC | DEF | FM | IPR | MB | OV | OPR | SV | IV | BC | LR |
|---|---|---|---|---|---|---|---|---|---|---|---|
| Aojmus | **0.878** | 0.900 | **0.830** | **0.916** | **0.911** | **0.753** | **0.895** | **0.992** | **0.906** | **0.892** | **0.996** |
| MSCF | 0.839 | **0.941** | 0.771 | 0.878 | 0.864 | 0.674 | 0.881 | 0.868 | 0.876 | 0.877 | 0.988 |
| Staple | 0.799 | 0.878 | 0.779 | 0.860 | 0.852 | 0.696 | 0.816 | 0.851 | 0.863 | 0.860 | 0.797 |
| fDSST | 0.722 | 0.777 | 0.771 | 0.799 | 0.838 | 0.544 | 0.759 | 0.796 | 0.876 | 0.891 | 0.731 |
| KCF | 0.722 | 0.813 | 0.744 | 0.818 | 0.832 | 0.604 | 0.803 | 0.805 | 0.863 | 0.882 | 0.785 |

Note:
The best results are in bold.

**Table 4 Comparison of success rate on different attributes.**

| Algorithms | OCC | DEF | FM | IPR | MB | OV | OPR | SV | IV | BC | LR |
|---|---|---|---|---|---|---|---|---|---|---|---|
| Aojmus | 0.732 | 0.727 | **0.794** | 0.762 | **0.853** | **0.795** | 0.718 | 0.651 | 0.684 | 0.744 | 0.662 |
| MSCF | **0.790** | **0.891** | 0.757 | **0.805** | 0.852 | 0.664 | **0.799** | **0.794** | **0.851** | **0.863** | **0.931** |
| Staple | 0.736 | 0.798 | 0.719 | 0.765 | 0.801 | 0.575 | 0.724 | 0.726 | 0.793 | 0.787 | 0.604 |
| fDSST | 0.656 | 0.710 | 0.753 | 0.733 | 0.801 | 0.550 | 0.678 | 0.715 | 0.815 | 0.827 | 0.705 |
| KCF | 0.599 | 0.661 | 0.681 | 0.662 | 0.783 | 0.620 | 0.628 | 0.520 | 0.644 | 0.750 | 0.285 |

Note:
The best results are in bold.

**Table 5 Comparison of tracking speed of our algorithm and others.**

| Algorithms | Frames per second (FPS) |
|---|---|
| Aojmus | 74.85 |
| MSCF | 16.01 |
| Staple | 9.28 |
| fDSST | 83.29 |
| KCF | 219.03 |

## Qualitative analysis

Like our previous work in *Wang et al. (2022)*, in order to better verify the advantages of the proposed algorithm, four groups of representative video sequences are selected for analysis, as shown in Fig. 7.

The video sequence of Bird2 contains OCC, DEF, FM, IPR, and OPR attributes. In frame 10, all the five tracking algorithms work properly. With the progress of tracking, the MSCF, fDSST, and KCF algorithms begin to show significant tracking drift at frame 51 affected by occlusion, deformation, and rotation problems. From frame 73, only the Aojmus and Staple can track accurately after the target (bird) flips.

The video sequence of Lemming contains IV, SV, OCC, FM, OPR, and OV attributes. From frame 10 to frame 370, as the target is not significantly affected by occlusion, fast motion and illumination changes, and the position of the target does not change after the occlusion, all the algorithms can track the target accurately. After the 370th frame, the occluded target reappears. As the tracking models of the other algorithms use fixed update

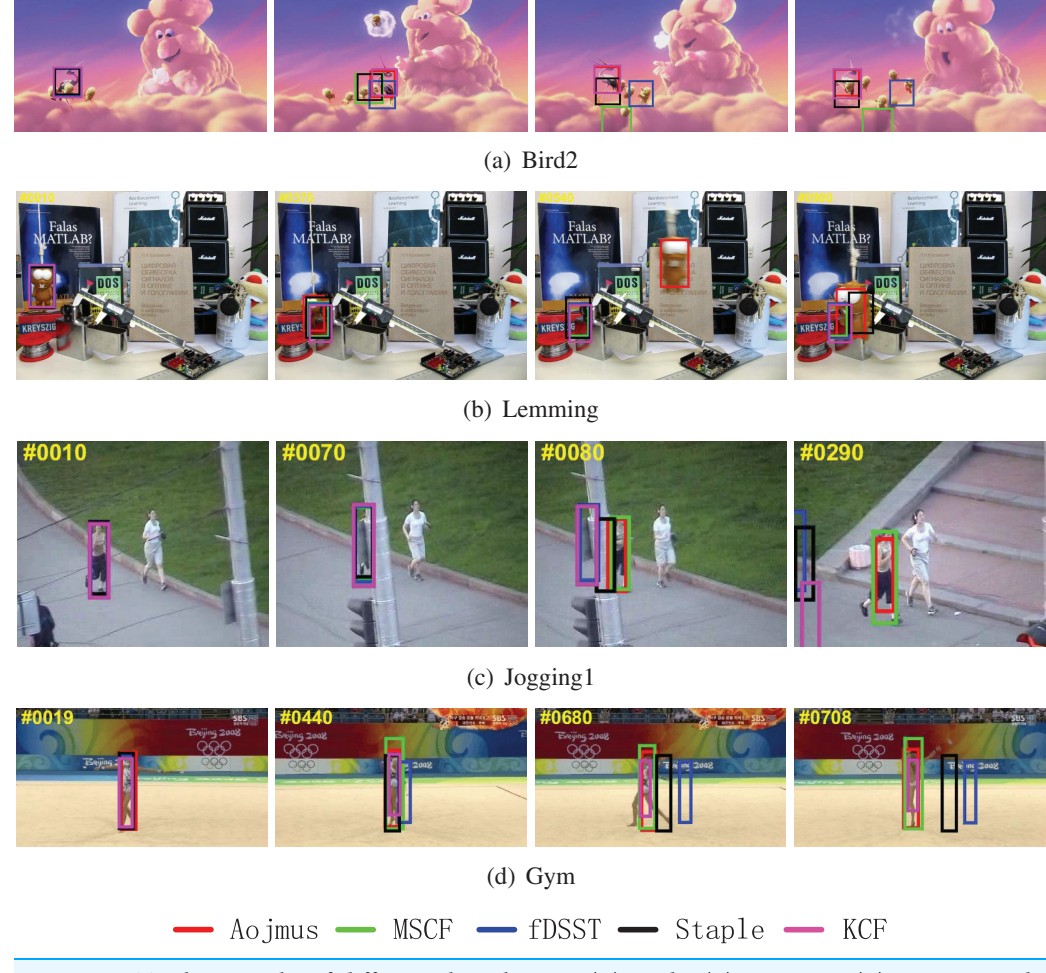

(a) Bird2

(b) Lemming

(c) Jogging1

(d) Gym

— Aojmus — MSCF — fDSST — Staple — KCF

**Figure 7 Tracking results of different algorithms on (A) Bird2, (B) Lemming, (C) Jogging1 and (D) Gym.** Photo credit: Visual Tracker Benchmark.

rate and learn non-target information, they are unable to locate the target again in the subsequent frames, while the Aojmus can maintain accurate localization until the end of tracking. In addition, comparing the tracking boxes at 900th frame, it can be shown the Aojmus is also well adaptive to the scale change of the target.

The video sequence of Jogging1 contains OCC, DEF, and OPR attributes. The target is heavily occluded at frames 70 to 80. When the target reappears, the Aojmus is able to accurately locate the target using adaptive occlusion judgment and continue the model update to ensure that tracking is performed reliably. The other algorithms except MSCF fail to cope with the occlusion problem.

The video sequence of Gym contains SV, DEF, IPR, and OPR attributes. At frame 19, the target begins to rotate and deform, and all algorithms can basically guarantee normal tracking. When it comes to 440th frame, the fDSST algorithm shows obvious drift, and fails to track the target. With the increase of target deformation and rotation, only the Aojmus, MSCF and KCF can keep tracking properly till frame 680. The Aojmus can

adjust the tracking box with the scale of target dynamically whereas the MSCF and KCF fail to do so.

The above experimental results show that the Aojmus proposed in this article can cope well with a variety of influence appearing in the tracking process, especially in the aspects of occlusion, scale change and deformation. On the whole, the Aojmus is robust for target tracking in complex scenes, and typically provides a new idea to deal with occlusion problems.

## CONCLUSIONS

In target tracking, scholars have conducted in-depth research in many aspects to be able to predict the position of moving targets more accurately. However, due to the variability of the tracked target and scene, it is not easy to develop an algorithm that takes into account the above 11 influencing factors simultaneously, especially in solving the problems of target occlusion, deformation and scale variation. The previous researches, which typically uses one judgment indicator to address the occlusion problem, can't obtain outstanding overall performance. In this study, considering the complex scenarios and the requirement of mutual-complementarity of technologies, we propose four indicators, $F_{max}$, $N$, $W_{APCE}$ and $RSFM$ as conditions to make the occlusion judgment more accurate. Moreover, we introduce an adaptive model updating strategy, fuse the results of the occlusion judgement and apply them into the model updating, which improves the precision in predicting the target position. As tracking is processed frame by frame where different influence factors may be encountered, this study presents a dynamic dual thresholds to compose the update strategy and achieves an accurate judgment of the existence and degree of occlusion, which solves the problem of tracking drift. In order to make full use of the feature information of target and reduce the influence of scale variation, we also incorporate a multi-feature fusion scheme and a scale estimation model in the backbone of the algorithm, which provides a good basis for later obscuration judgments and model updates.

The experimental results show that the Aojmus precedes the other typical tracking algorithms in terms of tracking precision, which has been increased by 0.6% and 3.8% respectively compared with the excellent algorithms, MSCF and Staple. Despite the Aojmus is not the best in terms of success rate, it surpasses the other four compared algorithms with respect to target occlusion, scale variation, fast motion, out-of-plane rotation and deformation. As the Aojmus is based on the kernel correlation filtering method, it runs well in real-time with high tracking speed of 74.85 frames per second, striking a good balance between tracking effectiveness and speed. It can be concluded that the kernel correlation filter-based multi-indicator occlusion judgement mechanism and adaptive model updating strategy can solve the common problems of target tracking while maintaining the overall performance. In future, we plan to investigate the feasibility of synthesizing our method with convolutional neural networks to improve the overall performance further and extend the application to indoor mobile robot and vehicle violation.

### Funding

This work is supported by the Natural Science Foundation of Fujian Province (No. 2018J01637), the Start-up Research Project of Fujian University of Technology (No. GY-Z21064), the New Engineering Research Project of Fujian University of Technology and the Research Project of Experimental Teaching Reform of Fujian University of Technology (No. SJ2018002). The funders had no role in study design, data collection and analysis, decision to publish, or preparation of the manuscript.

### Grant Disclosures

The following grant information was disclosed by the authors:
Natural Science Foundation of Fujian Province: 2018J01637.
Start-up Research Project of Fujian University of Technology: GY-Z21064.
New Engineering Research Project of Fujian University of Technology.
Research Project of Experimental Teaching Reform of Fujian University of Technology: SJ2018002.

### Competing Interests

The authors declare that they have no competing interests.

### Author Contributions

- Zhiming Cai conceived and designed the experiments, authored or reviewed drafts of the article, and approved the final draft.
- Zhuangzhuang Wang performed the experiments, performed the computation work, prepared figures and/or tables, authored or reviewed drafts of the article, and approved the final draft.
- Jianchao Huang analyzed the data, prepared figures and/or tables, and approved the final draft.
- Shujing Chen analyzed the data, prepared figures and/or tables, and approved the final draft.
- Huabin He performed the experiments, authored or reviewed drafts of the article, and approved the final draft.

### Data Availability

The public test dataset, OTB-2015 (namely TB-100) are available at Visual Tracker Benchmark: http://doi.org/10.6084/m9.figshare.24427468.

The experiment results underlying this article and the source codes are available at GitHub and Zenodo:

- https://github.com/fzxincai/Aojmus
- fzxincai. (2023). fzxincai/Aojmus: A target tracking method based on adaptive occlusion judgment and model updating strategy (Aojmus). Zenodo. https://doi.org/10.5281/zenodo.7563961.

## Supplemental Information

Supplemental information for this article can be found online at http://dx.doi.org/10.7717/peerj-cs.1562#supplemental-information.

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
