# Peer review of "A target tracking method based on adaptive occlusion judgment and model updating strategy"

_PeerJ Computer Science, doi:10.7717/peerj-cs.1562_

## Round 0.1 · original submission · Major Revisions

Based on reviewers comments, the authors are advised to make "major revisions".

Reviewer 1 ·

Basic reporting

The background and literature survey information can be added as another section in the manuscript.

Experimental design

no comment

Validity of the findings

no comment

Reviewer 2 ·

Basic reporting

This research presents A target tracking method based on adaptive occlusion judgment and model updating. However, the following suggestions are recommended:
• Abstract is written very poor. Extensive English mistakes are needed to address. Your abstract does not highlight the specifics of your research or findings.
• The introduction section is looking like related work or literature review. In the introduction section I suggest: problems, Aim, Methods, Results, and Conclusion. The author needs to explain the major factor of the manuscript. The paper's aim has been evidenced very poor, it is strongly suggested to highlight the originality and added value of the present work with respect to the Literature about the same topic. Introduction suffers from a lack of motivation and innovations. It should be expanded to include a more detailed discussion of current problems.
• The paper organization section is missing at the end of the introduction of section. Briefly describe the section and subsection of your whole menu script in one paragraph. Add this paragraph at the end of the introduction section.
• Related work is missing add related work section after introduction section
• Results and Discussion; the author should compare the finding of the present study with the previous study and justify for more clarity.
• A number of algorithms are available for such kind of research why author selected only Aojmus algorithm?
• Would you explicitly specify the novelty of your work? What progress against the most recent state-of-the-art similar studies was made?
• Conclusions should be amended to incorporate a broader discussion of the significance and potential application of this specific study.
• For qualitative analysis why author chose only 5 groups not more than 5?
• There should be no consecutive headings, add some text between two headings.
• English throughout the manuscript needs to be improved.
• I will not recommend this article for publication in this prestigious journal. Resubmission is allowed after suggested changes.

Experimental design

Author should argue biasedness of dataset being used.

Validity of the findings

A number of algorithms are available for such kind of research why author selected only Aojmus algorithm?

Additional comments

Paper can be accepted after addressing comments

Reviewer 3 ·

Basic reporting

The English is reasonable for the most part. The introduction contains little context around the applications of this type of tracking. While one could imagine applications of this technology, it would be helpful if the authors could point some of these applications out. The article structure is fine.

Regarding the relevant results, there are no statistics in this study at all. It is unclear to me whether the authors' method is actually a qualitative improvement or an actual improvement in performance. In addition, the meaning of "success" is defined in the paper, but it is unclear why this metric was chosen. Is it standard in the field.

I am not qualified to comment in detail on the mathematical derivations, but they appear to be standard in the field.

Experimental design

The research does appear to be within the scope of the journal. While the research question is adequately defined (i.e. to produce a target tracking method that overcomes issues with latency and occlusion problems), it is not at all clear why this is being done beyond the fact that there are limitations in the current tracking methods.

It is not at all clear why there are no statistics in this paper. The graphs and results in the tables may or may not be statistically significant. Statistics are necessary.

In addition, how were the tracking examples chosen? Is it because they are "practical" or in some cases "real world examples of cases where tracking would be important. What characteristics or parameters of the target motion are important? These details should be included in the paper in order to determine how well the results are likely to extend beyond the stimuli in the study.

Validity of the findings

At the moment there is nothing to indicate that these results are or are not important or significant without statistics. In addition, there is no discussion of how the stimuli were chosen and exactly what these stimuli consist of in terms of the boundaries of the occlusion, the movement in the scene, and the parameters of the targets themselves. Does the angular subtense of the objects matter (i.e. is something closer to the center of the screen more easily tracked)?

Additional comments

As written, the results in this paper are not compelling because there are insufficient details in the methods and analyses to draw conclusions as to whether there are truly significant improvements in tracking with the method the authors are developing. In addition, the closing statements in which the authors say that they will continue to develop the method seems odd, and makes this paper seem like an intermediate step toward something that works well.

---

## Round 0.2 · Major Revisions

The reviewers commented on the revised manuscript which requires "major revisions". The authors are advised to revise the manuscript accordingly and resubmit.

Reviewer 3 ·

Basic reporting

The changes to the paper have now reduced the readability. The grammar will need extensive repair before this paper could be ready for publication.

I still do not understand whether the improvements in performance are significant. The authors really need to give the reader some perspective on what one would consider improvement in tracking performance. As it reads currently, the technology looks promising (as many of the outcome measures are best with the authors' algorithm), but it's basically impossible to know if it is truly better. At minimum the authors should put in perspective what they think the results mean in terms of significance.

Experimental design

While the significance of the study is better defined in the revision, it's still not clear exactly how much improvement is provided with the new algorithm. Can the authors put the results in perspective for a real-world scenario?

Validity of the findings

There is better emphasis in the revision on the quantitative improvements, but there is still no clear indication of what would constitute a meaningful improvement for real-world applications.

---

## Round 0.3 · accepted · Accept

Dear Authors,
The paper is accepted, however, it is advised that comments from Reviewer 2 regarding English grammar must be solved when submitting final files. Here are the comments:

The grammar still needs some work. For example, in the section starting with line 127 there is this statement "Therefore, Gaussian kernel function j (xi) is introduced for linear transformation, and get..." The word "the" is missing, and the word "get" should be replaced with something like "and the result is.." In line 176 the word occlusion is spelled "cclusion".

Reviewer 2 ·

Basic reporting

See below

Experimental design

See below

Validity of the findings

See below

Additional comments

I have reviewed the manuscripts and the author has addressed all my concerns so I would like to accept this manuscript for publication in this journal.

Reviewer 3 ·

Basic reporting

The grammar still needs some work. For example, in the section starting with line 127 there is this statement "Therefore, Gaussian kernel function j (xi) is introduced for linear transformation, and get..." The word "the" is missing, and the word "get" should be replace with something like "and the result is.." In line 176 the word occlusion is spelled "cclusion".

Experimental design

This is fine.

Validity of the findings

Since there is no statistical comparison for the tracking performance for each method and since the authors do not state an accepted definition of "significant improvement", the conclusion that the Aojmus algorithm outperforms the other tracking algorithms cannot be fully supported.